# Effect of Chromium Content on the Oxidation Behavior of a Ta Stabilized γ’-Strengthened Polycrystalline Co-30Ni-10Al-4W-4Ti-2Ta Alloy

**DOI:** 10.3390/ma15175833

**Published:** 2022-08-24

**Authors:** Kun Qian, Shasha Qu, Lei Shu, Peng Xue, Xiaobing Li, Bo Chen, Kui Liu

**Affiliations:** 1Jihua Laboratory, Foshan 528200, China; 2Guangdong Institute of Special Equipment Inspection and Research, Guangzhou 510620, China

**Keywords:** CoNi-based superalloys, oxidation behaviors, parabolic growth

## Abstract

The high-temperature oxidation behaviors of polycrystalline Co-30Ni-10Al-4W-4Ti-2Ta superalloys with Cr contents ranging from 1 to 5 at.% are characterized at 900 °C to provide benchmark data for the alloy design of the CoNi-based superalloys. The mass gain curves for all three alloys exhibit parabolic growth, and the addition of 5Cr at.% is sufficient to decrease the oxidation rate by two orders of magnitude compared to the Cr-free alloy. Furthermore, cross-sectional findings reveal that these three alloys form qualitatively similar oxide scales composed of an outer oxide layer of Co_3_O_4_ and CoAl_2_O_4_ phase on top of an Al_2_O_3_ scale, following the inner oxide layers of Cr_2_O_3_, TiO_2_, and TiTaO_4_, and internally oxidized Al_2_O_3_ precipitate. The alloy forms a chromium-rich oxide scale as the Cr addition increased, and the concentration of Cr in the scale/alloy interface increases, promoting the growth of Cr_2_O_3_, while CoAl_2_O_4_ and Co_3_O_4_ nucleation is inhibited. The results further indicate that Cr has a superior effect on improving the oxidation resistance of CoNi-based alloys and that a higher content of Cr can assist the formation of a continuous Al_2_O_3_, Cr_2_O_3_, and TiTaO_4_ layers, which in turn hampers outer Co and Ni, and inward oxygen flux.

## 1. Introduction

The increasing demand for higher turbine entry temperatures (TET) to improve turbine efficiency pushes Ni-based superalloys to their operational limits [1]. As a result, a promising generation of Co-based superalloys is currently being developed to meet the aforementioned requirements. In 2006, Sato et al. [2] discovered a L1_2_ phase, Co_3_(Al, W), in the Co-Al-W ternary system. After that, a new Co-based alloy family was established through years of research, which has a similar strengthening mechanism to Ni-based superalloys [3,4,5,6,7,8,9,10]. However, the potential for (i) improving the creep behavior due to reduced coarsening caused by the low diffusivity of W in the matrix, (ii) reducing the freezing range, and thus improving castability, and (iii) higher stacking fault energies than Ni, and thus improved creep strengths, drive the development of these alloys [1]. The primary constraints for the development of Co-Al-W-based superalloys at the moment are lower γ’ solvus temperature (T_γ’_-solvus), weaker microstructural stability at high temperatures, and higher alloy density when compared to commercial Ni-based superalloys. Subsequently, some researchers have concentrated on addressing these shortcomings, and some of them found that Ni-containing Co-Al-W-based alloys are suitable for further development due to their higher T_γ’_-solvus and better microstructural stability than Ni-free counterparts [3,11]. Currently, with the development of such alloys, the discovery of coherent γ-γ’ microstructures in Co-Ni-Al-W-Ti-Ta systems opens up new opportunities for turbine engine materials [10,12,13]. Compared with primal Co-Al-W-based alloys, these alloys with the γ/γ’ dominant microstructure show superior high-temperature mechanical properties. In addition, the initial investigation shows that the alloy systems of Co-Ni-Al-W-Ti-Ta have a high T_γ’_-solvus and good microstructural stability and a low alloy density [10,11,14]. Thereinto, Qu et al. [13] have investigated the precipitation and coarsening behavior of the γ’ phase in the Co-30Ni-10Al-4W-4Ti-2Ta alloy and indicated that the Co-30Ni-10Al-4W-4Ti-2Ta alloy has a high γ’ solvus temperature and excellent microstructural stability at high temperature. Results also show that the B addition can strengthen the grain boundaries due to the formation of boride [6], and the addition of Ti and Ta can raise the γ’ solvus temperature. However, the oxidation behavior of the above alloy has not been systematically evaluated, especially for the effects of Cr addition. Generally, without the addition of Cr or Si, the oxidation resistance of Co-30Ni-10Al-based alloys is subpar [15]. The parabolic rate constants of the Co-30Ni-10Al-10W alloy after isothermal oxidation in air at 900 °C and 1000 °C, for example, are 0.00524 and 0.0302 mg^2^∙cm^−4^∙s^−1^, respectively [16]. It is well known that improving the high-temperature oxidation resistance of superalloys is also an important task of alloy design. The traditional cobalt-based superalloys have the best hot corrosion resistance in aggressive environments, in general. These alloys form continuous dense protective oxide scales, such as CoO, Cr_2_O_3_, SiO_2,_ and Al_2_O_3_. These oxides are thermodynamically more stable, growing more slowly, and more adherent to the surface compared to oxides formed on nickel-based superalloys. However, CoNi-based superalloys are known to present lower corrosion/oxidation resistance than that of Ni-based superalloys, which is the main limitation for their use in harsh environments at elevated temperatures. Therefore, the high-temperature oxidation behavior of the new CoNi-based alloys has to be improved considerably. So far, very little research is available for the CoNi-based superalloy systems. Zhuang et al. [17] have investigated the oxidation of Co-30Ni-(6-8)Al-3W-1Ta-(3-6)Ti-(12-14)Cr, and the effects of Al, Ti, and Cr were discussed. Li et al. [15] discussed the effect of Cr on the microstructural stability, oxidation property, and γ’ phase nano-hardness of multi-component CoNi-based superalloys. The effect of different amounts of Cr (4~8 at.%) on the oxidation resistance at 800 °C has also been investigated in a Co-9Al-9W-0.04B alloy by Klein et al. [18]. Moreover, Silva et al. [19] have concluded that the addition of Cr improved the alloy’s oxidation behavior by limiting the excessive formation of Co oxides and promoting the growth of protective oxides, such as Cr_2_O_3_ and Al_2_O_3_.

On the other hand, as previously stated, the oxidation resistance of the current experimental alloys is poor due to the lack of environmental resistance elements, such as Cr and Si. Cr is a well-known element for improving oxidation resistance by stabilizing a passive oxide layer. However, such additions can also hasten the formation of undesirable secondary phases. During high-temperature oxidation, most conventional Ni-based turbine alloys contain enough Cr to form a continuous, slow-growing, Cr_2_O_3_-rich scale, which protects the underlying alloy from further oxidative degradation and can self-heal if damaged. The addition of Cr to all Co-based superalloys to improve oxidation resistance is widely accepted. However, it has been discovered that the addition of Cr can affect a variety of properties, ranging from precipitate size to creep performance. Moreover, Ti is detrimental to the oxidation resistance due to the easy formation of TiO_2_ rutile and TiN particles. Yan and Dye [20] reported that Cr alloying destabilizes the γ-γ’ structure by forming topologically close-packed (TCP) phases, and the addition of Ni widens the γ-γ’ phase field, thereby restoring the γ-γ’ microstructure. Numerous studies have explored the oxidation resistance and Cr oxide formation mechanism in Co-based superalloys to identify an appropriate amount of Cr needed for improving oxidation resistance [18,20,21,22]. It is shown that Cr strongly distributes in the γ matrix, although it affects the γ’-solvus temperature and volume fraction of γ’ precipitates. Higher temperatures require excellent environmental degradation resistance in oxidizing environments to meet the operational service life of the component. However, the high-temperature oxidation behavior of this new class of materials has until now not been systematically investigated. Hence, the alloys studied here include 1, 3, and 5 at.% Cr, respectively.

The current study aims to evaluate the effect of Cr content on the high-temperature oxidation resistance of superalloys of the CoNi-based superalloy at 900 °C and to exposit the oxide scales’ characteristics in order to establish the oxidation sequence of different alloying elements and reveal the mechanism of oxidation, particularly to explore the role of chromium in the formation of continuous layers of Al_2_O_3_, Cr_2_O_3_, and TiO_2_ near the metal–oxide interface. The work intends to clarify the reported changes in oxidation processes that arise from the increasing Cr content in the Co-30Ni-10Al-4W-4Ti-2Ta alloys. These findings will serve as a baseline for the new γ’-strengthened CoNi-based superalloy design. Therefore, a deeper understanding of the aforementioned processes will be provided in order to improve the design of oxidation-resistant superalloys.

## 2. Materials and Methods

Gravity casting with high-purity raw materials in a vacuum induction furnace (BR-RLL-10, BROTHER FURNACE, Zhengzhou, China) produced the alloys. Table 1 shows the nominal and measured compositions of the alloys investigated in this study, with each alloy given an abbreviated label denoting the nominal atomic fractions of the various chromium-alloyed samples. Inductively coupled plasma optical emission spectroscopy (OPTIMA8300DV, PE, Waltham, MA, USA) was used to determine the actual compositions.

To eliminate any secondary phases, each ingot was a vacuum-sealed solution heat treated in a muffle furnace (KSL-1400X, KJ GROUP, Hefei, China) at 1250 °C for 24 h. Following that, all of the ingots were aged at 900 °C for 24 h before being removed from the furnace and cooled down to room temperature to achieve the desired γ-γ’ microstructure. For isothermal oxidation studies, 8.5 mm cylinders were cut from the ingot using wire cut-EDM (Electrical Discharge Machine) (GS1, Hanqi, Suzhou, China), and the surfaces were polished to a surface finish of 0.25 μm, cleaned, and degreased with an ultrasonic cleaning machine, and then dried with hot air. The samples were kept in the recrystallized alumina crucibles, which did not change in mass before or after the experiments. The crucibles were covered with alumina lids before weighing to accommodate spallation products during periodic specimen removal and air cooling. Moreover, for each oxidation temperature, a batch of three samples of each alloy was used to obtain reliable results. Each isothermal oxidation was carried out in the following manner: insert the specimens into the 900 °C furnaces; sample oxidation in the furnace environment for the allocated period; removal of the samples from the furnace at 900 °C. For about 30 min, the samples were cooled from 900 °C to room temperature. The mass change of the samples before and after the experiments was measured with an electro-balance (XS105, METTLER TOLEDO, Zurich, Switzerland) with a resolution of 10 μg, and a minimum of three samples was weighed for each data point, with the average value plotted. The mass change was measured after 1, 3, 5, 10, 25, 50, 75, and 100 h of exposure.

To minimize spallation during subsequent cross-sectional investigations, oxidized samples were protected by a chemically deposited Ni layer. Ni layers may detach due to stresses during the metallographic preparation of the cross-sections and thus are not always visible in the backscattered electron (BSE) micrograph. The surface, morphology, thickness, and structure of the oxide layers were examined using a Field Emission Gun Scanning Electron Microscope (Apreo, FEI, Hillsboro, OR, USA) with samples hot mounted in thermoset resin, ground, polished, and gold sputtered. Energy dispersive X-ray spectroscopy was used to measure the composition (EDS). Furthermore, the oxidized samples were subjected to X-ray diffraction (D/Max-2500PC, Rigaku Corporation, Tokyo, Japan) to identify the oxides and other phases, operating with Cu Kα radiation (45 kV, 200 mA), and the scanning angle ranged from 10° to 90°, and the scan step size was 0.012°. A scanning electron microscope was used to examine the surface morphologies and cross-sections of the oxide scales (ESEM Quanta 200, FEI, Hillsboro, USA). Image-Pro Plus software (Media Cybernetics, Inc., Rockville, MD, USA) was used to determine the volume fraction of the γ’ phase in the alloys [9] and to analyze the cross-sectional SEM images of oxide layers to determine the oxide layer thickness grown at 900 °C for 100 h.

## 3. Results

### 3.1. Alloy Microstructure and Constitution before Oxidation

The SEM images of the typical microstructure in Co-30Ni-10Al-4W-4Ti-2Ta-xCr (x = 0, 1, 3, and 5) alloys after the standard heat treatment are shown in Figure 1. It is worth noting that there is no precipitation of secondary phases in the microstructure of these alloys. Previously, it was discovered that the addition of chromium could reduce the solvus temperature of γ’, and similar phenomena are observed in this study [15,23], as shown in Table 1.

It can be seen that 0Cr, 1Cr, and 3Cr alloys have similar γ/γ’ matrix microstructures. The morphology of the γ’ phase varies from a cuboid shape (0Cr and1Cr alloys) to an irregular L shape (3Cr alloy) with increasing Cr content. However, the morphology of the γ’ phase for the 5Cr alloy takes on irregular shapes, which is clearly different from the first three alloys. Moreover, although the chromium content ranges from 0 to 5 at.%, the γ’ volume fractions are all about 80%. The addition of 1 and 3 at.% of chromium to these alloys under identical heat treatment conditions has no effect on the microstructures, as shown in Figure 1a–c, respectively, while the microstructure of the 5Cr alloy is obviously different from the first three alloys. The 1Cr, 3Cr, and 5Cr alloys have microstructures that are consistent with those reported by Park et al. [25] and Li et al. [14,15].

### 3.2. Oxidation Behavior

#### 3.2.1. Global Oxidation Kinetics

Figure 2 indicates the mass gain curves of the four alloys as a function of oxidation time after isothermal oxidation at 900 °C for up to 100 h [24]. The results reveal that increasing the Cr content improves the oxidation resistance of CoNi-based alloys at high temperatures and that the 5Cr alloy has better oxidation resistance than that of the other two alloys, resulting in a much thinner oxide layer. When oxidized at 900 °C, the Cr-containing alloys exhibit higher oxidation resistance and lower continuous mass gain compared to the Cr-free alloy.

It can also be observed that the mass gain (Δm/A) increases rapidly with oxidation time after about 1 h, which is called the transient period of oxidation [26,27]. The main characteristics of the transient period are that a thin surface layer of the alloy is converted into oxide with little diffusion occurring due to the rapid oxygen uptake by the alloy. The latter stage of the transient oxidation period is characterized by the development of layers of certain oxide phases in the scale. After a transient period, the growth of the oxide scales on the three alloys at 900 °C obeys the parabolic law for up to approximately 100 h, as shown in Figure 2.

At 900 °C, the plots of mass gain versus time all show a parabolic growth behavior. The increase in mass during oxidation is due not only to the formation of oxides but also to the enrichment of oxygen and nitrogen on the surface. A general power-law equation can be used to describe the evolution of mass gain with oxidation time:(1)$${\left(\Delta m\right)}^{n}=k_{p}\times t$$
where Δm is the mass gain per unit area, mg∙cm^−2^, $k_{p}$ is the oxidation rate constant, $mg^{n}\cdot cm^{-2n}\cdot s^{-n}$, *n* is the rate exponent, and *t* is the oxidation time, h. According to Wagner’s oxidation theory [28], when the oxide scale growth is a diffusion-controlled process, the *n* exponent is 2, and *k_p_* becomes the parabolic rate constant. The $k_{p}$ values thus identified in the parabolic rate range are summarized in Table 2.

#### 3.2.2. Morphology and Phase Composition of Oxidized Surfaces

Figure 3 shows a representative surface oxide structure observed after 100 h of oxidation at 900 °C in the air for all three alloys, revealing three distinct external scale morphologies. After high-temperature oxidation, the surface of the Co-30Ni-10Al-4W-4Ti-2Ta-xCr (x = 1, 3, and 5) superalloys is rough and relatively uniform, and the morphology of each oxide surface is distinctive. The images look morphologically similar to the scales after 100 h of oxidation, but the average size of the surface oxide particles decreases from 5.3 μm to 2.1 μm as the Cr content increases compared to that of the previous state.

The indexed XRD patterns of the surfaces of the three alloys studied are shown in Figure 4. There are different XRD patterns for three alloys after oxidation at 900 °C, indicating that these alloys may produce different types of oxidation products. CoAl_2_O_4_ and Co_3_O_4_ are the most visible phases in the XRD patterns of Co-30Ni-10Al-4W-4Ti-2Ta-xCr (x = 1, 3, and 5, at.%). Furthermore, the reflections of several additional oxide phases, such as rutile CrTaO_4_ [29,30,31] and TiN, are seen in the XRD patterns of 3Cr and 5Cr alloys at 900 °C.

#### 3.2.3. Cross-Sectional Investigation of Oxide Layers

Micrographs of cross-sections after 100 h exposure of Co-30Ni-10Al-4W-4Ti-2Ta-xCr alloys (x = 0, 1, 3, and 5 Cr at.%) at 900 °C are summarized in Figure 5. SEM images of the specimens oxidized in the air reveal clear differences in the oxide scale morphologies on the samples of the four materials. However, there is a very similar structure and a comparable adhesion of the oxide layers for all alloys, which consists primarily of three distinct oxide layers: the outer oxide layer, the inner oxide layer, and internal precipitation. On top of the original alloy surface, an outer oxide layer (d1) forms. The inner oxide layer is formed by a mixture of various oxide phases (d2). A region (d3) can also be seen between d2 and the unoxidized alloy, where granular or needle-like isolated phases that appear dark in the BSE micrograph precipitated in the unaffected matrix phase [29]. Compared to the Cr-free alloy in Figure 5a, the thicknesses of the oxide layer are much lower in the Cr-containing alloys; the oxide layer thickness decreases with the increase in chromium content, as shown in Figure 6, which is consistent with the change of the oxidation rate constant. The oxide layer of the Cr-free alloy is almost eight times thicker than that of the 5Cr alloy, whereas the difference in internal oxidation zones (d2 + d3) thickness is roughly 36 µm.

The analysis of oxidized samples’ microstructure in the cross-section shows differences in the oxidation behavior of these four kinds of CoNi-based superalloys. The results reveal that the 5Cr alloy shows the best oxidation resistance due to the formation of compact oxide scales. Therefore, the formation of protective spinel oxide scales is conducted with increasing chromium addition, which favors better oxidation resistance.

#### 3.2.4. Details on Multilayered Oxide Scale Growth

The elemental composition and appearance of the cross-sections are studied after isothermal oxidation by using scanning electron microscopy (SEM) and an energy dispersive spectrometer (EDS). After 100 h of exposure, the elemental mappings of regions in samples 1Cr, 3Cr, and 5Cr that are affected by the formation of oxide scales are shown in Figure 7, Figure 8 and Figure 9, respectively. For the 1Cr alloy, the elemental mappings of the cross-sections after 100 h exposure are summarized in Figure 7. After about 100 h of exposure, an outer oxide layer forms on top of the original alloy surface, containing primarily cobalt oxides, such as CoAl_2_O_4_ and Co_3_O_4_. Furthermore, at 900 °C, a continuous and protective Cr_2_O_3_ layer forms, resulting in the formation of a discontinuous Al_2_O_3_ layer with no comparable protective effect. It can also be observed that long-term oxidation of the 1Cr alloy results in the formation of a significant internal oxidation particle layer with plate-like alumina. Following the Al profile, one can observe an enrichment of Al in the inner part of the oxide scale, which corresponds to the internal precipitates of Al_2_O_3_.

The enrichment of the Al element at the oxide/alloy interface promotes the formation of a protective Al_2_O_3_ oxide layer during the scale growth, which is preferable to slow the rate of growth [1]. Moreover, the d3 region of dilution in elements can be observed in the figure where there are isolated phases that appear dark in the EDS mappings precipitate on top of the unaffected matrix phase [29]. The formation of alumina particles causes Al depletion from the bulk material, which results in the dissolution of the γ’ phase and a local weakening of the alloys [30].

Figure 8 shows the SEM micrographs and EDS mappings of the 3Cr alloy oxidized at 900 °C. Again, the cross-sections of the 3Cr alloy oxidized samples present the formation of a complex multilayer of oxides. This sample’s oxide scales can be classified into three distinct regions, including an outermost layer of Co_3_O_4_ and TiN, a middle oxidation zone containing continuous Al_2_O_3_, TiO_2_, and TiTaO_4_ layer, and an internal oxidation zone of dispersing precipitates of Al_2_O_3_. The formation of the TiTaO_4_ phase was already observed by Zhuang et al. [17] and Gao et al. [31]. Furthermore, there are some white precipitates in the middle oxidation. Compared to the middle oxidation zone in Figure 7, the addition of Cr alloying elements increases the density and stability of the transition layer in the middle of the oxide film. The zone of Cr enrichment is relatively wide, which suggests that the majority of the oxide scale consists of Cr_2_O_3_.

Figure 9 shows EDS mappings of the 5Cr alloy oxidized at 900 °C for 100 h. It can be observed that the proportion of the outer oxide layer in the total thickness of the oxide layer is significantly reduced. An amount of 5 at.% Cr alloying for the Co-30Ni-10Al-4W-4Ti-2Ta alloy produces a protective, 9.1 μm thick scale; the outer is Co_3_O_4_, CoAl_2_O_4,_ and CrTaO_4_, and the inner is a porous TiO_2_, TiTaO_4_, and Al_2_O_3_ mixture. Therefore, there are protective oxide scales. Additionally, white precipitates Co_3_W are found in the outer oxide layer due to the removal of Al [20]. Compared to the discontinuous thin Al_2_O_3_ layer formed in the 1Cr alloy (in Figure 7), the layer formed in the 3Cr and 5Cr alloys reveals a continuous structure in the intermediate oxide layer. Additionally, an inner, virtually continuous alumina precipitates below the TiTaO_4_ scale, which causes aluminum depletion. Elongated precipitates are again observed between the γ/γ’ matrix and the Al_2_O_3_-rich region in the innermost discontinuous layer, which is still granular Al_2_O_3_, and TiO_2_ seems to serve as a nucleation site for further alumina formation and growth in the inner oxide layer. Moreover, compared to the γ’-depleted zone in Figure 7 and Figure 8, the oxidation of the 5Cr alloy results in a less continuous γ’-depleted region adjacent to the Al_2_O_3_-rich zone. Therefore, the findings show that Cr_2_O_3_ appears to have a positive effect on the formation of continuous Al_2_O_3_ and TiTaO_4_ layers.

In addition, there are some Ti-rich phases in the Al_2_O_3_-rich zone. It is inferred that TiO_2_ promotes the formation of alumina as a nucleating particle. In the present study, the 3Cr and 5Cr alloys form the Cr_2_O_3_ and Al_2_O_3_ layer and exhibit a mass gain of 0.0025 and 0.0014 mg/cm^2^ after isothermal oxidation at 900 °C for 100 h (Table 2), respectively, demonstrating significantly higher oxidation resistance than the 0Cr alloy. Therefore, increasing the Cr content of the 5Cr alloy results in a similar but thinner oxide layer than the 3Cr alloy (Figure 7), indicating improved oxidation performance at 900 °C. Moreover, Al_2_O_3_ grows at a slower rate than Cr_2_O_3_ and is non-volatile. Because Cr_2_O_3_ evaporates at temperatures above 900 °C, the Cr_2_O_3_ layers in both the 3Cr alloy and the 5Cr alloy are porous (Figure 7) and could not provide adequate oxidation protection for the substrate. Alloys containing 3 and 5 at.% Cr exhibit a flatter, more compact external scale and Cr_2_O_3_ layer, as well as reduce oxygen ingress, after 100 h of oxidation at 900 °C.

Figure 10 plots a linear analysis of chemical composition in the cross-section after oxidation. It can be observed that the alumina layer is located above the Ti oxides, followed by the chromium oxide layer, and the concentration change of Ta is consistent with Ti in some locations. Some results show that the alloy containing Ti and Ta forms a TiTaO_4_ layer compound between TiO_2_ and internal oxide Al_2_O_3_, and TiTaO_4_ is rutile-type oxide [32]. A continuous TiTaO_4_ oxides layer is observed beneath the Cr_2_O_3_ layers in both the 3Cr and 5Cr alloys, and they are more likely to form above the discontinuous Al_2_O_3_ particles [17], indicating that the continuous alumina layer and chromium oxide layer contribute to the formation of the TiTaO_4_ layer. Meanwhile, the formation of a continuous TiTaO_4_ layer on top of the dense Cr_2_O_3_ layer can improve oxidation resistance even further.

## 4. Discussion

### 4.1. The Evolution of Oxide Film Structure

In this study, the formation of diffusion-limiting barrier layers, as well as the extent of sub-scale phase transitions, are found to be directly related to the alloy’s Cr content. At the start of exposure, Co-oxide formation occurs at the alloy surface, resulting in a Co-depleted zone in the matrix and subsequent Al-, Ta-, Cr-, and Ti enrichment in these Co-depleted areas, while oxygen diffuses inward, causing internal oxidation. Previous research has shown that adding Cr significantly expands the γ/γ’ two-phase region to the aluminum-rich side, increasing the γ/γ’ two-phase region’s Al solubility.

In general, a continuous Al_2_O_3_ layer forms only when the Al concentration in the alloy system exceeds a critical value, whereas internal alumina particles, such as those found in the 1Cr alloy, form when the Al concentration is less than this value. A sufficiently high thermodynamic flux will result in a more stable scale former, in this case, Cr, Ta, or Al, which must be sustained from the adjacent regions of the alloy. In other words, at a given oxygen flux, the diffusion velocities of Al and Cr within the alloy are critical. Several investigations determined the diffusion coefficients for Ti, Al, Ta, Mo, Cr, and W in pure binary Co alloys; the order of the diffusion coefficients in fcc-Co is as follows: ${\bar{D}}_{\left(Ti\right)}>{\bar{D}}_{\left(Al\right)}>{\bar{D}}_{\left(Ta\right)}>{\bar{D}}_{\left(Mo\right)}>{\bar{D}}_{\left(Cr\right)}>{\bar{D}}_{\left(W\right)}$ [33]. The aforementioned trend is also seen in EDS line scans across the Cr-depleted and Al-depleted regions, as shown in Figure 10. From the thermodynamics, due to the lower free energy of the formation, Al_2_O_3_ should be the first oxide to form, and the next are oxides of tantalum and titanium, as illustrated in Figure 11. Therefore, the alumina layer is firstly formed near the d1/d2 interface, and studies have shown that the addition of Cr reduces the critical Al concentration value for the formation of a continuous Al_2_O_3_ layer. Because the growth of these Al_2_O_3_ precipitates consumes a significant amount of Al, lowering the oxygen’s partial pressure in this region, Cr can diffuse to near the d1/d2 interface and eventually form a continuous layer, which is located below the Al_2_O_3_ layer. At the same time, titanium oxides also begin to form near the d1/d2 interface, and the change in the concentration of the Ti element is always immediately after the Al element, as shown in Figure 11. The reason that the continuous TiTaO_4_ layer lies behind the TiO_2_ layer can be attributed to the slow diffusion rate of the Ta atom and the higher diffusion rate of the Ti atom. Moreover, the primary TiO_2_ provides nucleation sites for the tantalum oxide, resulting in the formation of TiTaO_4_ [34].

Moreover, the parabolic rate constant values for the 3Cr and 5Cr alloys show that the Al_2_O_3_ and Cr_2_O_3_ layers can slow the surface reaction rate by delaying the outward diffusion of Co/Ni ions and the inward diffusion of O ions. As a result, the external oxide scale grows slowly. Due to this, the activity of oxygen near the oxide substrate interface decreases more than in the Cr-free alloy. The transport of oxygen through d1 toward the d2/d3 interface drives the formation of an internal oxidation zone. The oxygen partial pressure decreases with the increase in Cr content, and the diffusion of Ta atoms toward the d2/d3 interface is suppressed. Meanwhile, the transient TiO_2_ provides nucleation sites for Al_2_O_3_ at the internal oxidation zone, which promotes the formation of granular alumina, when the Al content is below a critical value in the alloy system. The higher Cr concentration in the 5Cr alloy allows the formation of dense continuous Al_2_O_3_ and Cr_2_O_3_ layers (as shown in Figure 8). The overall higher Cr content in the depleted zone of the 5Cr alloy indicates that the diffusion-limiting layer has more effective barrier properties [29], as shown in Figure 10c. Therefore, the thickness of the oxide layer decreases with the increase in Cr concentration.

In conclusion, the addition of 3Cr and 5Cr in the present alloys is conducive to the formation of the Al_2_O_3_, TiTaO_4,_ and Cr_2_O_3_ oxide layers, whereas the presence of Cr in Ti- and Ta-containing Co-30Ni-10Al-4W-4Ti-2Ta alloys slows the oxidation kinetics and influences the oxide scale morphology.

### 4.2. Composition of Multilayered Oxides after 100 h Exposure

Figure 12 shows the relationship between the parabolic rate constants and reciprocal temperature of the Co-Ni-Al-W-based alloy systems, Co-Ni-Al-Mo-based alloy systems, and some nickel-based superalloys at 700–1000 °C using data from this study and available data from the literature. It is clear that the three alloys investigated in this study have typical Cr_2_O_3_ oxidation kinetics, despite the fact that they also form alumina layers. The parabolic constants determined in this study are lower than those determined by Yeh et al. [16] for the oxidation of Co-30Ni-10Al-10W at 900 °C and Li et al. [15] for the oxidation of Co-30Ni-5Cr-9Al-6W-4Ti-1Ta at 1000 °C. It should be noted that the growth of the oxide layer on the 5Cr alloy is significantly slower than that on the 3Cr and 1Cr alloys, which is primarily governed by the growth of Cr_2_O_3_ and Al_2_O_3_. At 900 °C, the oxidation resistance of 5Cr is comparable to that of conventional nickel-based alloys, such as RR1000, but differs from that of typical CoNi-based superalloys, such as Co-30Ni-10Al-10W, as shown in Figure 12.

As reported in previous investigations, all Co-based γ/γ’ alloys with a typical amount of aluminum (approximately 10% at.%) have a low oxidation resistance [19]. However, the addition of Cr significantly improves the oxidation resistance, demonstrating the synergistic effect of Al and Cr on the formation of a continuous Al_2_O_3_ layer former [15]. On the other hand, the oxidation behavior of the alloys in this study is quite similar to that of the CoNi-based alloy (Co-30Ni-5Cr-11Al-4W-4Ti-1Ta) studied by Li et al. [15], which has a higher Cr concentration (5Cr-9Cr, at.%) and lower Al and Ta concentration. So far, no type of Co-based γ/γ’ alloy has entered the area of the alumina layer.

Chromium has a significant impact on the oxidation behavior of all alloys. As previously stated, chromium alloying encourages the formation of continuous crystalline Cr_2_O_3_ in both 3Cr and 5Cr alloys. Meanwhile, a thick alumina barrier layer forms behind a two-phase mixed outer oxide layer of TiO_2_ and TiTaO_4_ in the 5Cr alloy. Although the chromium oxide layer is very effective at suppressing oxide growth, a Cr_2_O_3_ layer will not provide adequate protection above 900 °C, so a continuous Al_2_O_3_ layer plays a role as the protective film, and this is the target and direction of the alloy optimization.

## 5. Conclusions

A comparative study of the high-temperature oxidation behavior of the new γ’-strengthened Co-30Ni-10Al-4W-4Ti-2Ta-xCr (x = 1, 3, and 5, in at.%) is conducted to better understand the prevailing oxidation mechanisms, as well as the development and composition of multilayered oxide scales by measuring a more complete determination of mass change during isothermal exposure in the air, and the scale growth behavior in the previously introduced series of Cr-alloying CoNi-based alloys exposure is elucidated. The following conclusions can be drawn:(1)The oxidation kinetics of the three alloys is parabolic at 900 °C for durations up to 100 h, with the parabolic oxidation rate decreasing as the chromium content increased. Furthermore, long-term oxidation results in the formation of granular alumina.(2)Under isothermal oxidation, three layers of oxides are discovered in the 3Cr and 5Cr alloys. The outer layer is composed of Co_3_O_4_, CrTaO_4_, and CoAl_2_O_4_, while the middle layers are composed of Al_2_O_3_, Cr_2_O_3_, TiO_2_, and TiTaO_4_, and the internal oxides are Al_2_O_3_ precipitates. These low-oxygen diffusivity oxide scales adhere well to the substrate and each other, protecting the alloy from high-temperature oxidation. Nonetheless, the effect of Cr on the formation of Cr_2_O_3_ is the most important factor in improving oxidation resistance.(3)In both 3Cr and 5Cr alloys, oxidation-resistant and protective oxide scales can form, and samples with higher Cr contents demonstrate greater resistance to scale growth. At 900 °C for 100 h, 5% Cr and 10% Al result in multilayer oxide scales of Al_2_O_3_, Cr_2_O_3_, and TiTaO_4_ that are protective and limit oxygen inward diffusion. The three alloys investigated in this study have typical Cr_2_O_3_ oxidation kinetics, even though they also form alumina layers.(4)Using the oxide parabolic growth rate as a guide, adding 5 at.% Cr reduces the oxidation rate by three orders of magnitude. Without the addition of Cr, Al alone cannot form a protective oxide scale on the surface at 900 °C. Therefore, chromium may aid in the formation of stable Cr_2_O_3_, Al_2_O_3_, and TiTaO_4_ layers, which in turn hampers outer Co and Ni, and inward oxygen flux.

## Figures and Tables

**Figure 1 materials-15-05833-f001:**
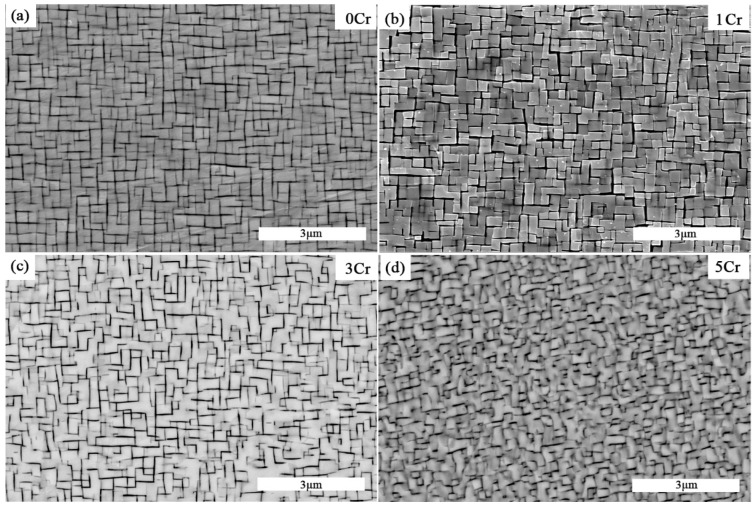
The SEM images of microstructures of the Co-30Ni-10Al-4W-4Ti-2Ta alloys with different Cr content after homogenization heat treatment at 1250 °C for 24 h and annealing at 900 °C for 24 h (**a**) 0Cr [24]; (**b**) 1Cr; (**c**) 3Cr; (**d**) 5Cr.

**Figure 2 materials-15-05833-f002:**
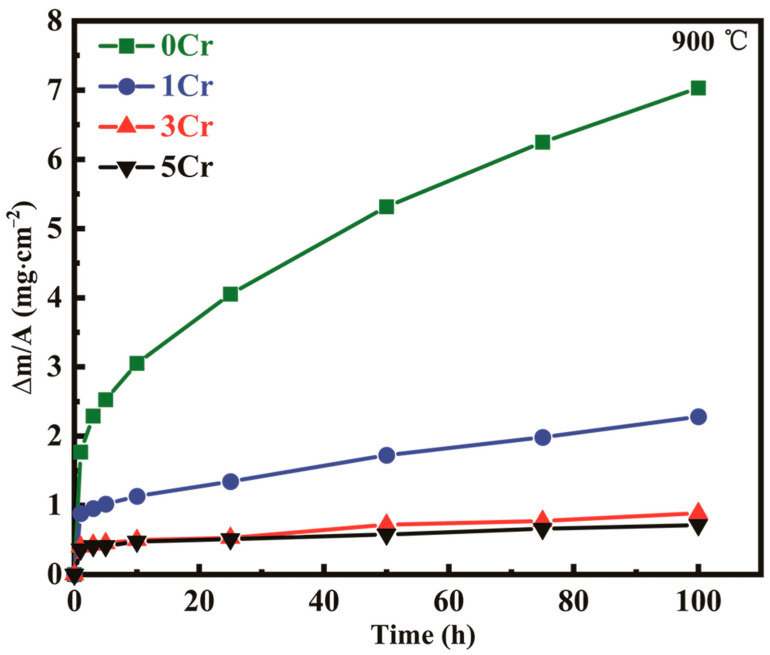
Evolution of mass gain (Δm/A) with oxidation time of CoNi-based alloys (oxidized for 100 h).

**Figure 3 materials-15-05833-f003:**
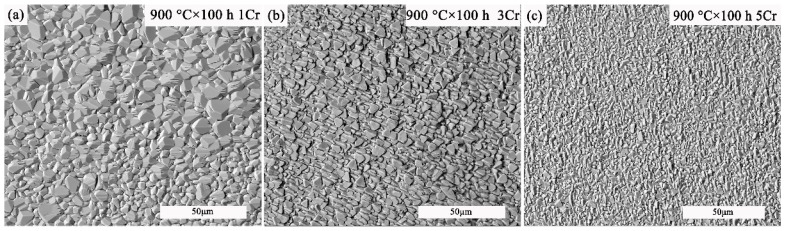
SEM micrograph of oxidized surface, (**a**) 1Cr 900 °C/100 h, (**b**) 3Cr 900 °C/100 h, (**c**) 5Cr 900 °C/100 h.

**Figure 4 materials-15-05833-f004:**
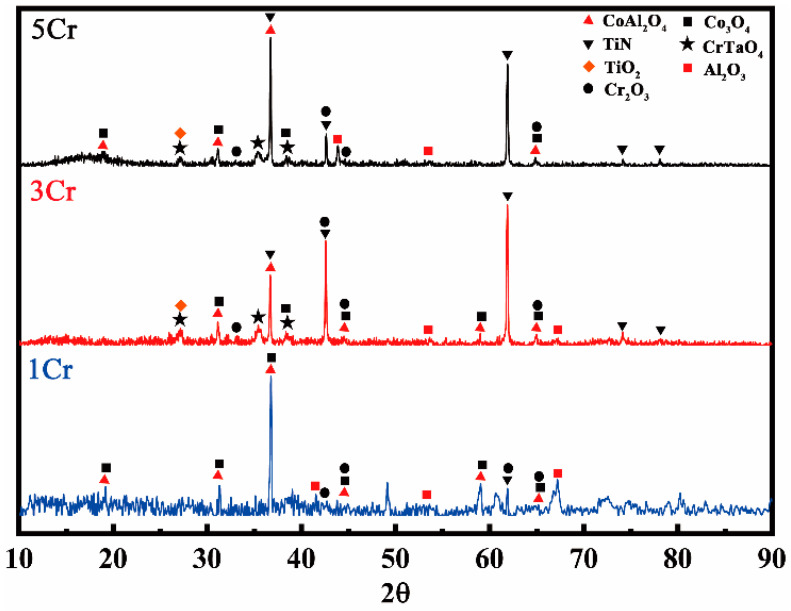
X-ray diffraction patterns of surface indexed to the oxide phases grown after oxidizing for 100 h at 900 °C.

**Figure 5 materials-15-05833-f005:**
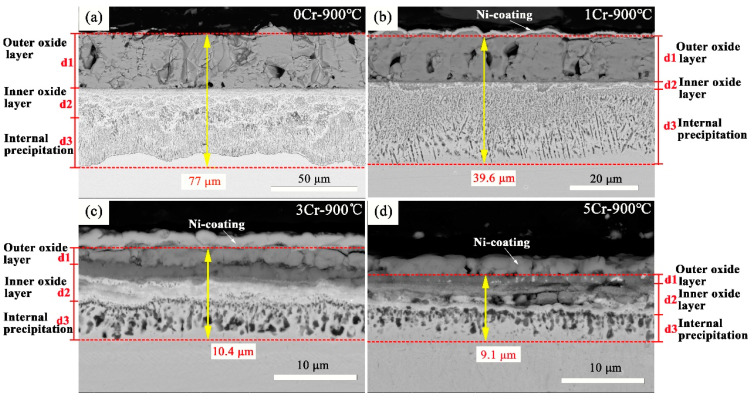
BSE micrographs of multilayered scales on CoNi-based alloys after 100 h exposure at 900 °C in air, (**a**) 0Cr [24]; (**b**) 1Cr; (**c**) 3Cr; (**d**) 5Cr.

**Figure 6 materials-15-05833-f006:**
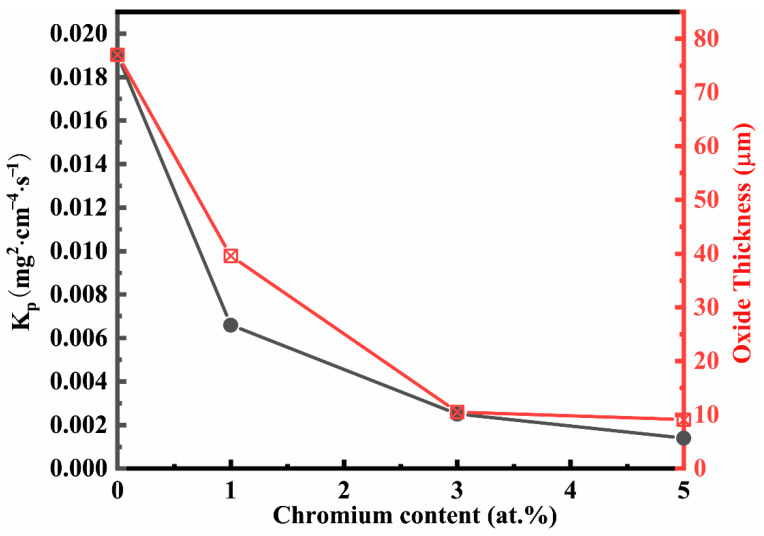
The plot of oxide scale thickness and parabolic rate constant as a function of Cr concentration for the four alloys at 900 °C for 100 h is shown in Figure 5, and the red font is the oxide thickness.

**Figure 7 materials-15-05833-f007:**
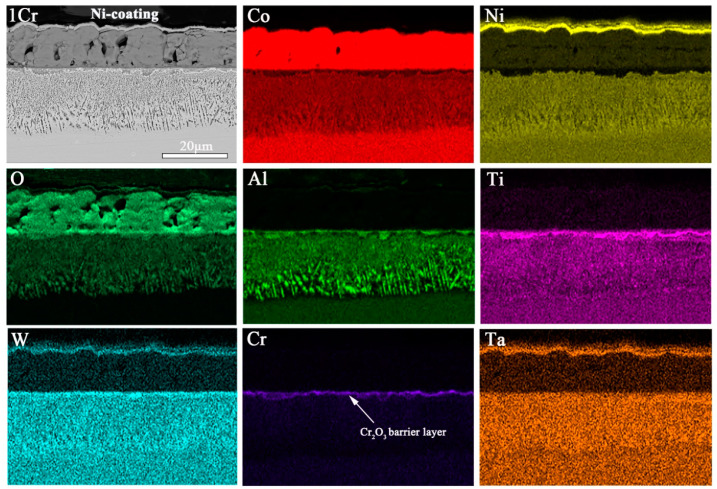
SEM micrographs and EDS mappings of Co-30Ni-10Al-4W-4Ti-2Ta-1Cr alloy oxidized at 900 °C for 100 h.

**Figure 8 materials-15-05833-f008:**
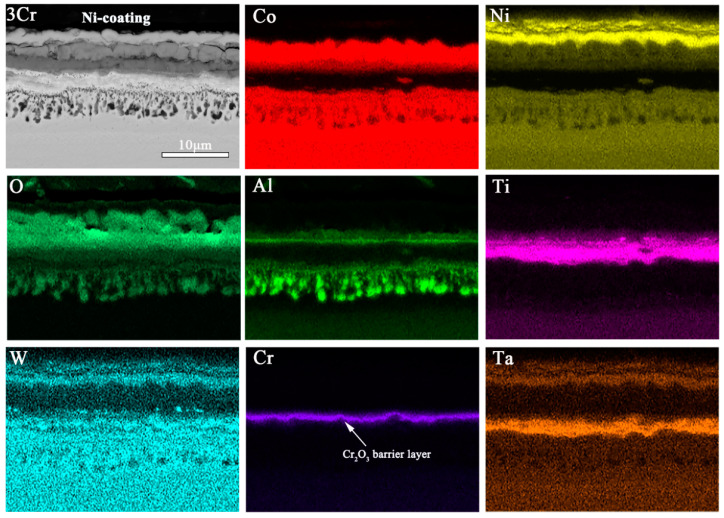
SEM micrographs and EDS mappings of Co-30Ni-10Al-4W-4Ti-2Ta-3Cr alloy oxidized at 900 °C for 100 h.

**Figure 9 materials-15-05833-f009:**
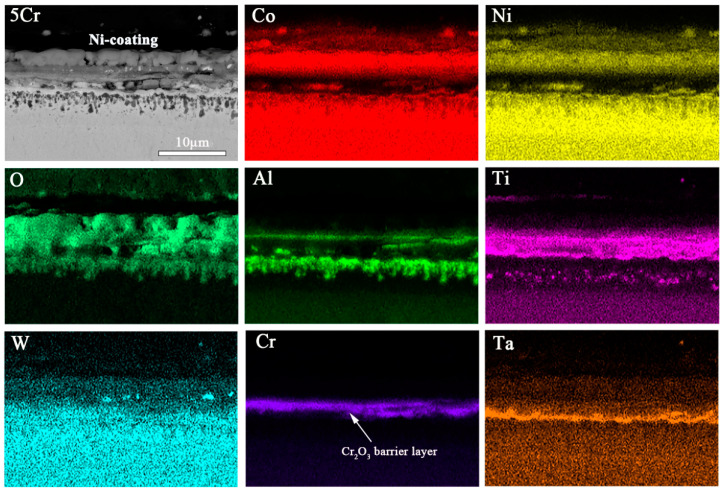
SEM micrographs and EDS mappings of Co-30Ni-10Al-4W-4Ti-2Ta-5Cr alloy oxidized at 900 °C for 100 h.

**Figure 10 materials-15-05833-f010:**
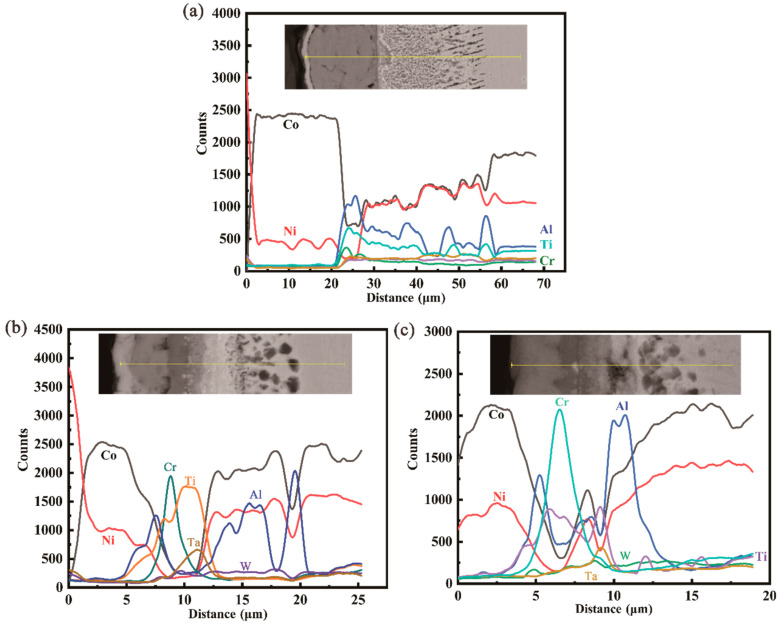
Linear analysis of chemical composition in cross-section after isothermal oxidation at 900 °C for 100h: (**a**) Co-30Ni-10Al-4W-4Ti-2Ta-1Cr; (**b**) Co-30Ni-10Al-4W-4Ti-2Ta-3Cr; (**c**) Co-30Ni-10Al-4W-4Ti-2Ta-5Cr.

**Figure 11 materials-15-05833-f011:**
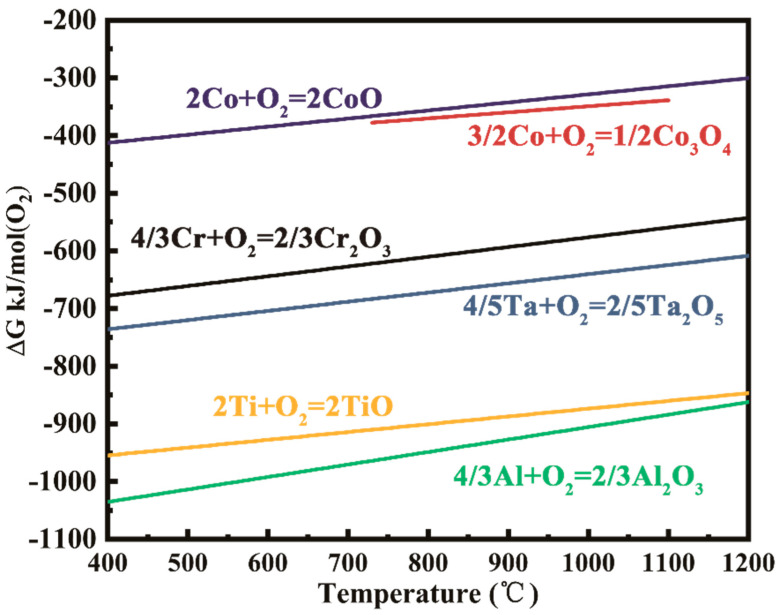
The relationship between the formation of Gibbs free energy of various oxides materials and temperature.

**Figure 12 materials-15-05833-f012:**
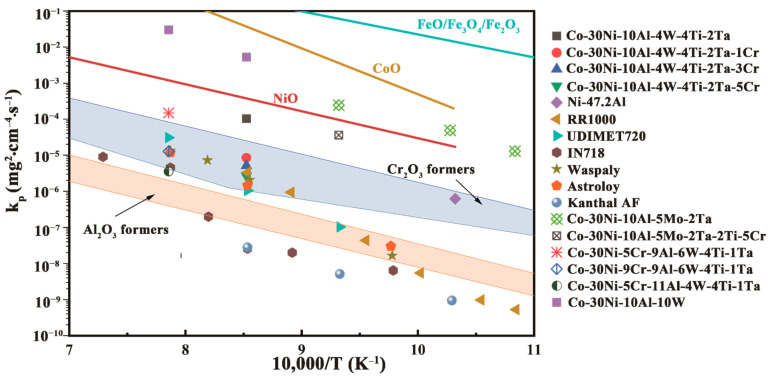
Parabolic rate constants versus reciprocal temperature for some Ni- and Co-based alloys from the literature and the values determined in this study [15,23,35]. The shaded areas represent the typical values of chromium and alumina formers. Adapted from Refs [19,36].

**Table 1 materials-15-05833-t001:** Chemical composition (in at.%) of the investigated CoNi-based alloys and utilized abbreviations.

Alloy Designation	Nominal Composition	Measured Composition								T_γ’s_ (°C)
		Co	Ni	Al	W	Ti	Cr	Ta	B	
1Cr	Co-30Ni-10Al-4W-4Ti-2Ta-1Cr	Bal.	30.27	9.21	4.04	3.96	0.99	2.01	0.26	1213
3Cr	Co-30Ni-10Al-4W-4Ti-2Ta-3Cr	Bal.	30.40	9.24	4.02	3.95	2.94	2.01	0.27	1202
5Cr	Co-30Ni-10Al-4W-4Ti-2Ta-5Cr	Bal.	30.54	9.35	4.02	3.97	4.96	2.02	0.24	1191

**Table 2 materials-15-05833-t002:** Parabolic rate constant (*k_p_*) for all the alloys at 900 °C.

Alloy Designation	*k_p_* (mg^2^∙cm^−4^∙s^−1^)	*n*
0Cr	1.03 × 10^−4^	2
1Cr	2.66 × 10^−6^	2
3Cr	5.12 × 10^−6^	2
5Cr	8.58 × 10^−6^	2

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
