# Peer review of "Effect of Chromium Content on the Oxidation Behavior of a Ta Stabilized γ’-Strengthened Polycrystalline Co-30Ni-10Al-4W-4Ti-2Ta Alloy"

_materials, 2022, doi:10.3390/ma15175833_

Round 1

Reviewer 1 Report

The abstract has been well written; the main results, obtained in the work, have been provided.

The introduction has been well written. A sufficient amount of work has been presented. The goal has been written specifically and it is consistent with the results of the work. The methods have been described comprehensibly. Sample preparation before the studies has been provided. The authors use reasonable methods and approaches to the study of the oxide layer. Data on the changes in the mass of samples, width of the oxide layer, etc. are given. In the “Discussion” section, the authors compare the obtained data with the previously published results of other authors. The authors have formulated clear-cut and easy-to-understand conclusions.

However, there are some comments/questions and recommendations:

(1) Authors ought to check the number of words. According to the template, the abstract should contain no more than 200 words.

(2) Line 113: the authors write “... a deeper understanding of the enhanced processes will be provided in order to improve the design of oxidation-resistant super alloys”. Have the authors managed to formulate an approach to the development of the mentioned alloy, based on the conclusions? It would be of interest if the authors formulated minimum requirements, based on the obtained results, for example, chromium concentration, annealing time, flake shape, etc. This is a recommendation.

(3) I also recommend that the authors in Figure 2 and Figure 4 use the same color for the same composition. For example, according to Figure 2, “5Cr” is blue, and the same composition is already black in Figure 4. The same color palette will facilitate the perception of the data.

(4) Line 223: Is it possible to provide data on the size of the particles formed on the surface, on the direction of the scales' growth? Is this data of importance to your work?

(5) In Figure 3a, one can see that larger particles are represented in the photomicrograph from the middle upwards. What is the reason for this?

(6) There is no explanation of the ‘BSE’ abbreviation (line 145)

(7) The caption to Figure 5 is highlighted in bold.

(8) The table caption format varies.

Author Response

Dear reviewers:

Thank you very much for your consideration, and we really appreciate the comments and have learned a lot. Appropriate changes were made and highlighted in the revised manuscript according to the suggestions of reviewers and editor.

The abstract has been well written; the main results, obtained in the work, have been provided.

The introduction has been well written. A sufficient amount of work has been presented. The goal has been written specifically and it is consistent with the results of the work. The methods have been described comprehensibly. Sample preparation before the studies has been provided. The authors use reasonable methods and approaches to the study of the oxide layer. Data on the changes in the mass of samples, width of the oxide layer, etc. are given. In the “Discussion” section, the authors compare the obtained data with the previously published results of other authors. The authors have formulated clear-cut and easy-to-understand conclusions.

However, there are some comments/questions and recommendations:

(1) Authors ought to check the number of words. According to the template, the abstract should contain no more than 200 words.

Response: According to your advice, we have condensed the abstract, which has less than 200 words.

(2) Line 113: the authors write “... a deeper understanding of the enhanced processes will be provided in order to improve the design of oxidation-resistant superalloys”. Have the authors managed to formulate an approach to the development of the mentioned alloy, based on the conclusions? It would be of interest if the authors formulated minimum requirements, based on the obtained results, for example, chromium concentration, annealing time, flake shape, etc. This is a recommendation.

Response: Thanks for your advice. Using the oxide parabolic growth rate as a guide, adding 5 at.% Cr reduces the oxidation rate by three orders of magnitude at 900 °C, which shows the best oxidation resistance.

(3) I also recommend that the authors in Figure 2 and Figure 4 use the same color for the same composition. For example, according to Figure 2, “5Cr” is blue, and the same composition is already black in Figure 4. The same color palette will facilitate the perception of the data.

Response: According to your advice, we have adjusted the same color palette in Figures 2 and 4.

(4) Line 223: Is it possible to provide data on the size of the particles formed on the surface, on the direction of the scales' growth? Is this data of importance to your work?

Response: According to your advice, we have measured the size of particles formed on the surface, and the data on the particle’s size has been added to the manuscript, such as ‘The images look morphologically similar to the scales after 100 h of oxidation, but the average size of the surface oxide particles decreases from 5.3μm to 2.1μm as the Cr content increases, compared to that of the previous state.

(5) In Figure 3a, one can see that larger particles are represented in the photomicrograph from the middle upwards. What is the reason for this?

Response: The size of the surface oxide is not uniform, especially grain boundaries are fast channels for diffusion of elements, resulting in more rapid nucleation and growth of surface oxide. Therefore, larger particles are represented in the photomicrograph from the middle upwards.

(6) There is no explanation of the ‘BSE’ abbreviation (line 145)

Response: we have added the explanation of the ‘BSE’ abbreviation in the manuscript, such as ‘the backscattered electron (BSE) micrograph’.

(7) The caption to Figure 5 is highlighted in bold.

Response: We are sorry for our mistakes and have corrected the caption to Figure 5.

(8) The table caption format varies.

Response: According to your advice, we have adjusted the table caption in the manuscript.

Reviewer 2 Report

This manuscript is well-written and easy to read. It presents interesting results on high temperature corrosion (in Air) of Co-30Ni-10Al-4W-3 4Ti-2Ta Alloy, and the effects of Cr addition on the corrosion behaviour. Overall, I think the paper can be published in the present for with a few small changes: 

Experimental:
- the specifications of ICP-OES is missing?
-the details of XRD measurements is missing. 

in Results: 

- oxidation behaviour: line 181-182: Is there any evidence for the layer thickness? ( SEM images of the cross-section? literature? ) 

- For EDX figures: please mention the time of oxidation test? ( 100h, true?) 

One small note: the introduction is comprehensive and summarises the history of these alloys. It clearly mentions the issue and the aim of study; however, the first paragraph seems to be a rather lengthy. ( it is a personal opinion, though).

Author Response

Dear reviewers:

Thank you very much for your consideration, and we really appreciate the comments and have learned a lot. Appropriate changes were made and highlighted in the revised manuscript according to the suggestions of reviewers and editor.

  1. Experimental: the specifications of ICP-OES is missing? the details of XRD measurements is missing. 

Response: According to your advice, we have added the specifications of ICP-OES and the details of XRD measurements in the manuscript, such as “Inductively coupled plasma optical emission spectroscopy (OPTIMA8300DV, PE, Massachusetts, USA) was used to determine the actual compositions” and “the oxidized samples were subjected to X-ray diffraction (D/Max-2500PC, Rigaku Corporation, Tokyo, JPN) to identify oxides and other phases, operating with Cu Kα radiation (45kV, 200 mA), and the scanning angle ranged from 10° to 90°, and the scan step size was 0.012°”

  1. in Results: 

① oxidation behaviour: line 181-182: Is there any evidence for the layer thickness? ( SEM images of the cross-section? literature? ) 

Response: Thanks for your advice. The size of the samples are same in this study, the less oxidation mass gain usually leads to lower oxide layer thickness, which is the circumstantial evidence of the oxide layer thickness. As you said, SEM images of the cross-section in Fig. 5 are direct evidence of the oxide layer thickness.

②For EDX figures: please mention the time of oxidation test? (100h, true?) 

Response: Thanks for your advice, and we have added the time of the oxidation test in the manuscript.

One small note: the introduction is comprehensive and summarises the history of these alloys. It clearly mentions the issue and the aim of study; however, the first paragraph seems to be a rather lengthy. ( it is a personal opinion, though).

Response: According to your advice, the first paragraph has been revised.

Reviewer 3 Report

The authors present a very interesting study on the Effect of chromium content on the oxidation behavior of a Ta 2 stabilized γ′-strengthened polycrystalline Co-30Ni-10Al-4W-3 4Ti-2Ta alloy. However, the article has the following problems that must be resolved before publication:

1.    In the Introduction, it is stated that Co-Al-W base superalloys at the moment have weaker microstructural stability at high temperatures when compared to commercial Ni-base superalloys.However, in the same section (line 64) the authors claim that the traditional cobalt-based superalloys have the best hot corrosion resistance in aggressive environments in general. Please explain this apparent contradiction. 

2.    In the Materials and Methods, nothing is said about how oxidation of the samples was performed. Later on, in the Results section, it is mentioned that it was in air (static ?).

3.    In Results section it is said (line 167) that the morphology of the γ′ phase varies from a cuboid shape (0Cr and1Cr alloys) to an irregular shape (3Cr and 5Cr alloy) with increasing Cr content. By varying the chromium content range from 0 to 5 at.%, γ′-volume fractions for the considered alloys are all about 0.80.  (how this value was calculated?) Moreover, it is said that the addition of 1 and 3 at.% of chromium to these alloys under identical heat treatment conditions has no effect on the microstructures as shown in Fig.1a-c, respectively, while the microstructure of 5Cr alloy is obviously different from the first two alloys. Please explain this apparent contradiction.

4.    Figure 2 caption refers Dm / A but in y axes the authors just write Dm.

5.    In line 197, it is written that the mass gain curve slope drops which means that the oxidation rate slows down after about 50 h of exposure.This is not evident from the shape of the curves.

6.    Why the XRD pattern of the 0%Cr sample is not shown? How these XRD patterns were obtained? Bragg-Brentano mode? Grazing incidence? Radiation?, etc…

7.    In line 252 the authors claim that the difference in internal oxidation zones thickness is roughly 36 μm. Please explain how this value was calculated.

8.    Line 257: inner Al2O3-formation is demoted, which leads to superior oxidation properties compared to Cr-free alloys. How do the authors know about this oxide from the SEM images?

9.    Why the EDS elemental mappings of 0Cr sample are not included in the paper? Why the N map was not acquired? This would support the formation of TiN.

10.  In the analysis of Figures 7 to 9 the authors claim the formation of phases that were not detected by XRD (e.g. Al2O3). Are these phases amorphous? If so, how do the authors know their chemical stoichiometry?

11.  TiN is not mentioned in the analysis of Figure 9. However, this phase was clearly detected by XRD (Please explain). Contrarily, the authors mention the formation of the Co3W phase which was not detected by XRD. (please explain). This is an important issue in this paper once the authors talk about phases and their stoichiometry based on the EDS elemental maps with no support from the XRD analysis. TEM technique should be used to verify the formation of these phases.    

12.  Fig. 10. represents EDS results and should be moved to the Results section. Moreover, once again the results for the 0%Cr sample are not shown.

13. In the paper, there are several repetitions of the same idea (e.g. lines 195 to 199, After a transient period, the growth of the oxide scales  on the three alloys at 900°C obeys the parabolic law for up to approximately 100 h, as  shown in Fig. 2. In addition, the mass gain curve slope drops which means that the  oxidation rate slows down after about 50 h of exposure. At 900 °C, the plots of mass gain vs. time all show a parabolic growth behavior.

14.  Finally, English should be improved.

Author Response

Dear reviewers:

Thank you very much for your consideration, and we really appreciate the comments and have learned a lot. Appropriate changes were made and highlighted in the revised manuscript according to the suggestions of reviewers and editor.

  1. In the Introduction, it is stated that Co-Al-W base superalloys at the moment have weaker microstructural stability at high temperatures when compared to commercial Ni-base superalloys. However, in the same section (line 64) the authors claim that the traditional cobalt-based superalloys have the best hot corrosion resistance in aggressive environments in general. Please explain this apparent contradiction.

Response: The traditional cobalt-based superalloys have the best hot corrosion resistance in aggressive environments due to the high chromium content, such as Stellite X-40 alloy, the chromium content is up to about 25 wt.%. However, Co-Al-W base superalloys are under development, and most of them do not contain chromium, so CoNi-based superalloys present lower corrosion/oxidation resistance than that of Ni-based superalloys and the traditional cobalt-based superalloys.

  1. In the Materials and Methods, nothing is said about how oxidation of the samples was performed. Later on, in the Results section, it is mentioned that it was in air (static ?).

Response: According to your advice, the detailed experimental process has been added to the manuscript, such as “The samples were kept in the recrystallized alumina crucibles which did not change in mass before or after the experiments. The crucibles were covered with alumina lids before weighing to accommodate spallation products during periodic specimen removal and air cooling. Moreover, for each oxidation temperature, a batch of three samples of each alloy was used to obtain reliable results. Each isothermal oxidation was carried out in the following manner: Insert the specimens into the 900°C furnaces in the air, and sample oxidation in the furnace environment for the allocated period, subsequently, remove the samples from the furnace at 900°C. For about 30 minutes, the samples were cooled from 900°C to room temperature. The mass change of the samples before and after the experiments was measured with an electron balance (XS105, METTLER TOLEDO, Zurich, CH) with a resolution of 10 μg, and a minimum of three samples was weighed for each data point, with the average value plotted. The mass change was measured after 1, 3, 5, 10, 25, 50, 75, and 100 h of exposure.”

  1. In Results section it is said (line 167) that the morphology of the γ′ phase varies from a cuboid shape (0Cr and1Cr alloys) to an irregular shape (3Cr and 5Cr alloy) with increasing Cr content. By varying the chromium content range from 0 to 5 at.%, γ′-volume fractions for the considered alloys are all about 0.80. (how this value was calculated?) Moreover, it is said that the addition of 1 and 3 at.% of chromium to these alloys under identical heat treatment conditions has no effect on the microstructures as shown in Fig.1a-c, respectively, while the microstructure of 5Cr alloy is obviously different from the first two alloys. Please explain this apparent contradiction.

Response: We are sorry for our mistakes, as you said, the microstructure of 5Cr alloy is obviously different from the first two alloys, and we have corrected the statement, such as “It can be seen that 0Cr, 1Cr, and 3Cr alloys have similar γ/γ′-matrix microstructures. The morphology of the γ′ phase varies from a cuboid shape (0Cr and1Cr alloys) to an irregular L shape (3Cr alloy) with increasing Cr content. However, the morphology of the γ′ phase for the 5Cr alloy takes on irregular shapes, which is clearly different from the first three alloys”. On the other hand, the volume fraction of the γʹ phase was calculated by the measured total area of the γʹ precipitates in SEM micrographs and the total area of the SEM micrographs with Image-Pro Plus software. To ensure accuracy, at least ten SEM micrographs were used to calculate these values. During the process of imaging segmentation, the histogram-based method and the “RGB” color mode were used, and the “R” button was selected. The maximum of the “R” value was 255, while the minimum was changed to ensure that almost all of the γʹ precipitates in one SEM micrograph could be selected. For imaging analysis, the “Filter Ranges” of the area were set from 10 to 10,000,000. The total area of SEM micrographs, Sγʹ, was calculated using ImageJ software. The total number of the γʹ precipitates in SEM micrographs, nγʹ, was counted manually. Then, the mean edge length of the γʹ precipitates was calculated by . The mean radius of the γʹ precipitates is expressed as half of the mean edge length. The γʹ number density, NV(t), was determined by the formula of NV(t)= NA(t)/a, where NA(t) is the total number of the γʹ precipitates per unit area and a is the mean edge length of γʹ precipitates. The detailed analysis process was referred to Reference[1], which has been added in the manuscript.

  1. Figure 2 caption refers Dm / A but in y axes the authors just write Dm.

Response: Thank you for pointing out the mistake, and we have corrected the Fig.2 caption.

Fig. 2 Evolution of mass gain ( ) with oxidation time of CoNi-based alloys (oxidized for 100 h)

  1. In line 197, it is written that the mass gain curve slope drops which means that the oxidation rate slows down after about 50 h of exposure. This is not evident from the shape of the curves.

Response: According to your advice, we have deleted the related statement.

  1. Why the XRD pattern of the 0%Cr sample is not shown? How these XRD patterns were obtained? Bragg-Brentano mode? Grazing incidence? Radiation?, etc…

Response: Because the result of the XRD pattern of the 0%Cr sample has been reported by ref.[2], and the XRD patterns of the 0%Cr sample is not shown. Moreover, the details of XRD measurements are added in the manuscript, such as “Inductively coupled plasma optical emission spectroscopy (OPTIMA8300DV, PE, Massachusetts, USA) was used to determine the actual compositions” and “the oxidized samples were subjected to X-ray diffraction (D/Max-2500PC, Rigaku Corporation, Tokyo, JPN) to identify oxides and other phases, operating with Cu Kα radiation (45kV, 200 mA), and the scanning angle ranged from 10° to 90°, and the scan step size was 0.012°”

  1. In line 252 the authors claim that the difference in internal oxidation zones thickness is roughly 36 μm. Please explain how this value was calculated.

Response: According to the thickness of d2+d3 region in the Fig.5a and 5b, we count the size of three different images of each alloy, and the average value was calculated.

  1. Line 257: inner Al2O3-formation is demoted, which leads to superior oxidation properties compared to Cr-free alloys. How do the authors know about this oxide from the SEM images?

Response: Thank you for pointing out the mistake, we have deleted the statement.

  1. Why the EDS elemental mappings of 0Cr sample are not included in the paper? Why the N map was not acquired? This would support the formation of TiN.

Response: Because the result for the 0%Cr sample has been reported by ref.[1]. In general, there is no N map for cobalt-based alloys oxidation studies[3-4]. Moreover, the ability of EDS to distinguish nitrogen elements is weak, as shown in Fig. 1, and the result also does not come close to supporting the formation of TiN, so there is no N map.

  1. In the analysis of Figures 7 to 9 the authors claim the formation of phases that were not detected by XRD (e.g. Al2O3). Are these phases amorphous? If so, how do the authors know their chemical

Response: The X-ray diffraction intensity is a statistical result of the sample surface information, and some of these phases in this study are determined by EDS and references [4-5]. Moreover, some phases were determined by re-analysis of XRD patterns, such as Al2O3, as shown in Fig.4.

  1. TiN is not mentioned in the analysis of Figure 9. However, this phase was clearly detected by XRD (Please explain). Contrarily, the authors mention the formation of the Co3W phase which was not detected by XRD. (please explain). This is an important issue in this paper once the authors talk about phases and their stoichiometry based on the EDS elemental maps with no support from the XRD analysis. TEM technique should be used to verify the formation of these phases.

Response: As I mentioned above, the X ray diffraction intensity is a statistical result of the sample surface information, and some of these phases are determined by EDS and reference [4-5]. Moreover, some phases were determined by re-analysis of XRD patterns.

  1. Fig. 10. represents EDS results and should be moved to the Results section. Moreover, once again the results for the 0%Cr sample are not shown.

Response: According to your advice, the EDS results have been moved to the Results section, and the result for the 0%Cr sample has been reported by ref.[2].

  1. In the paper, there are several repetitions of the same idea (e.g. lines 195 to 199, After a transient period, the growth of the oxide scales on the three alloys at 900°C obeys the parabolic law for up to approximately 100 h, as shown in Fig. 2. In addition, the mass gain curve slope drops which means that the oxidation rate slows down after about 50 h of exposure. At 900 °C, the plots of mass gain vs. time all show a parabolic growth behavior.

Response: Thanks for your advice, we have revised the relevant statement in the original manuscript.

  1. Finally, English should be improved.

Response: Thanks for your advice, we tried our best to improve the manuscript and made some changes to the manuscript. These changes will not influence the content and framework of the paper. And here we did not list the changes but marked in red in the revised paper. We appreciate for Editors/Reviewers’ warm work earnestly and hope that the correction will meet with approval.

References

  1. S. Qu, Y. Li, C. Wang, X. Liu, K. Qian, J. Ruan, Y. Chen and Y. Yang, Materials Science and Engineering: A, 787, 139455 (2020). https://doi.org/https://doi.org/10.1016/j.msea.2020.139455
  2. S. Qu, (University of Science and Technology of China: Hefei, China, 2021).
  3. P. Šulhánek, M. Drienovský, I. Černičková, L. Ďuriška, R. Skaudžius, Ž. Gerhátová and M. Palcut, Materials, 13 (2020). https://doi.org/10.3390/ma13143152
  4. W. Li, L. Li, S. Antonov, F. Lu and Q. Feng, Journal of Alloys and Compounds, 826, 154182 (2020). https://doi.org/https://doi.org/10.1016/j.jallcom.2020.154182
  5. (a) S.M. Das, M.P. Singh and K. Chattopadhyay, Corrosion Science, 172, 108683 (2020). https://doi.org/https://doi.org/10.1016/j.corsci.2020.108683; (b) M. Weiser, M.C. Galetz, H.-E. Zschau, C.H. Zenk, S. Neumeier, M. Göken and S. Virtanen, Corrosion Science, 156, 84-95 (2019). https://doi.org/https://doi.org/10.1016/j.corsci.2019.05.007; (c) B. Gao, L. Wang, Y. Liu, X. Song, S.Y. Yang and A. Chiba, Corrosion Science, 157, 109-115 (2019). https://doi.org/https://doi.org/10.1016/j.corsci.2019.05.036
